# Ecological niche modelling of two water-dependant birds informs the conservation needs of riverine ecosystems outside protected area network in the Eastern Himalaya, India

**Roshan Tamang**[1☯], **Vallanattu James Jins**[1☯], **Sailendra Dewan**[1], **Shivaji Chaudhry**[2], **Seema Rawat**[3], **Bhoj Kumar Acharya**[1]*

**1** Ecology, Biogeography and Conservation Biology Laboratory, Department of Zoology, Sikkim University, Tadong, Gangtok, East Sikkim, India, **2** Department of Environmental Science, Indira Gandhi National Tribal University, Amarkantak, Madhya Pradesh, India, **3** School of Life Sciences, Central University of Gujarat, Gandhinagar, Gujarat, India

☯ These authors contributed equally to this work.
* bkacharya@cus.ac.in, acharya2skm@gmail.com

**Data Availability Statement:** All the data are available as supplementary materials.

**Funding:** This study is funded by Ministry of Environment and Forests under National Mission

## Abstract

Common species often play vital roles in ecosystem functions and processes. Globally, conservation strategies are mostly focused on threatened species and rarely explored the potential of using common species as indicators of critical ecosystems. The Himalayan mountains have unique riverine ecosystems harbouring high diversity of specialist river birds. Ecological niche modelling provides effective tools to predict suitable habitats of a species and identify habitats for conservation. We used two common water-dependent bird species, Blue Whistling Thrush and White-capped Water Redstart as indicators of riverine ecosystems within the Sikkim Himalayan region and predicted their suitable habitats using an ensemble modelling approach. We selected six predictor variables for the final model including three bioclimatic and three topographic variables. For both species, bioclimatic variables such as mean annual temperature and precipitation were the most important factors compared to topographic variables. At least 70 percent of the most suitable habitats are distributed below 2000 m elevation alongside major drainages. Also, most of their potential habitats are distributed outside the protected area networks in the region. This habitat suitability pattern may be applied to other sympatric species in the region. Since major water bodies in Sikkim are largely affected by developmental activities and climate change, these riverine birds might face threats of losing suitable habitats. We recommend a dynamic approach to evaluate the habitat quality of riverine birds, especially outside protected area networks in the region to plan conservation strategies. This approach will ensure habitat conservation of many water-dependent birds and other taxa associated with the riverine ecosystems of the Eastern Himalaya.

on Himalayan Studies (Grant No: NMHS-2017/MG-01/477). The funding has been received by BKA.

**Competing interests:** The authors have declared that no competing interests exist.

## Introduction

The changes in land-use since the past five decades is reported to be one of the major threats to the terrestrial and freshwater ecosystems around the globe [1, 2]. It has been estimated that land-use and associated changes has led to a decline in local richness by 13.6%, and total abundance by 10.7%, globally [3]. The negative impact of land-use change on biodiversity is further projected to be exacerbated by global climate change. In order to safeguard biodiversity, it has become important to formulate conservation measures that expand across local, regional and international borders. Generating information on distribution limits and potential habitats of species are the key initial steps that are required for strategic conservation planning [4].

The availability of species occurrence and climatic datasets at local and global scale has provided more opportunities for deploying evidence-based conservation measures with the help of different modelling approaches. Species distribution models integrate occurrence records and environmental data to characterise bioclimatic conditions suitable for a particular species and then identify the most suitable habitats in their occurrence range [5, 6]. The recent advancements in distribution modelling have allowed conservation scientists to develop progressively more accurate models from both presence-only and presence-absence data of species [7, 8]. The outputs of such models are crucial for habitat conservation of target species and address the impacts of global changes [9, 10].

The mountain ecosystem, such as the Himalaya, remains one of the most sensitive regions to global changes. The rising temperatures (faster than the global average), rapid changes in land-use to meet up demands of increasing population causes serious threats to biodiversity in the Himalaya [11]. In particular, the riverine habitats in the low elevational zones and species therein are subjected to extreme pressure from construction of dams, establishment of industries, mining and expansion of road network [12]. Given the higher anthropogenic pressure and less protected area coverage, the lowland riverine ecosystem needs immediate attention for conservation [13].

Conserving common species is critical in retaining their key ecological and functional roles in ecosystems. Conservation strategies often focus on threatened species; however, we need a holistic approach to preserve multiple species and ecosystem functions. For instance, a habitat conservation approach focusing on the riverine ecosystem could benefit multiple species including sympatric birds and other water-dependent taxa such as dragonflies, damselflies, fishes, amphibians and reptiles. Such approaches are critical to inform local and regional decision-makers on conservation strategies that anticipate the response of biodiversity to future climate and land use changes. Owing to high sensitivity to habitat and climate, birds can be considered as the best vertebrate models and indicators to study the impacts of such changes [14]. River birds, in particular, can act as excellent cost-effective indicators of the human influences on riverine ecosystems in the Himalayan region [15].

Water dependent birds such as Blue Whistling Thrush *Myophonus caeruleus* and White-capped Water Redstart *Phoenicurus leucocephalus* are the most commonly occurring representatives of the riverine bird communities in the Himalaya and are indicators of pristine riverine ecosystems. These birds are conspicuous, fairly common and are easily censused, making them appropriate model species for distribution studies [16]. In the Himalaya, these birds share habitats with many other riverine birds including Plumbeous Water Redstart, Brown Dipper, Forktails and Wagtails [15, 17].

In this study, we modelled the distribution of Blue Whistling Thrush (hereafter BWT) and White-capped Water Redstart (hereafter WWR) in the Sikkim-Himalayan region with the aim of identifying suitable habitats that can be considered pristine areas and prioritized for conservation. The state of Sikkim is being impacted by large-scale changes and resulting threats to

the riverine ecosystems in the region. One-third of the state is covered by protected area (PA) networks which have a great role in protecting the natural habitats of many species [18]. However, the disproportionate distribution of PAs along the elevation and the rapid land-use changes in the region necessitate the identification of potential areas for conservation even outside PAs [19, 20]. Six hydroelectric projects have been proposed along the Teesta River of which two of them (Teesta stage-III and stage-V) have already been constructed and become operational [21]. Additionally, other developmental activities such as the construction of roads, bridges, and industries (pharmaceuticals factories) have also been taking place along different riverine ecosystems in the region. All these activities contribute to large-scale changes and threaten important biodiversity areas and riverine ecosystems of the Eastern Himalaya. Hence, the outputs of this study could be utilized in developing conservation strategies and policy guidelines for multiple species and dwindling riverine ecosystems in the Eastern Himalaya.

## Materials and methods

### Study area

We conducted this study in the Eastern Himalaya covering Sikkim state of India which lies between 27˚ 03' to 28˚ 07' N and 88˚ 03' to 88˚ 57' E (Fig 1). Despite its smaller size, Sikkim harbours high avifaunal diversity because of its geographical location, topography, elevational range and variety of vegetation types [22, 23]. The vegetation of Sikkim is categorized into three major types–Tropical, Temperate and Alpine [23]. The rainfall in Sikkim varies between 2000 to 5000 mm and is considered to be the wettest region in the entire Himalayan belt where the relative humidity is 70–80% throughout the year [24]. The climate varies along the elevation from hot tropical at the lower elevation to cool temperate at the middle and arctic cold at the higher elevation [25].

The drainage of the state is controlled by the perennial Teesta and Rangeet rivers along with their tributaries. Teesta (which forms the major drainage system) originates from the central crystalline zone defined by high mountain ranges covered by glaciers [26]. These rivers along with many other wetlands across the state have been hosts to many conservation concern bird species [27, 28]. By providing the critical habitat and breeding ground for threatened species, these wetlands also support diverse provisioning services for local communities [29].

### Species occurrence data

To obtain locality records of study species (Fig 2), we conducted multiple field surveys in different parts of Sikkim between 2008 and 2018. We covered all possible elevation zones and habitat types during the survey. We collected a total of 52 occurrence records of BWT and 40 records of WWR. The permission to conduct field work was granted by the Forest and Environment Department, Government of Sikkim, India. Since this study did not involve collection and capturing of animals, ethics approval was not necessary.

### Data preparation and distribution modelling

We downloaded bioclimatic variables for the study area from the Chelsa climatic database (https://www.chelsa-climate.org/), which is a fine-scale (i.e., 1 × 1 km), long-term (1979–2013) climate dataset with global coverage based on statistical downscaling [30]. Compared to other freely available datasets, the Chelsa climate is reported to be more effective for modelling species distributions in geographically complex regions such as the Himalaya [31]. Apart from bioclimatic variables, we also included elevation, slope, land-use and land cover (LULC), and

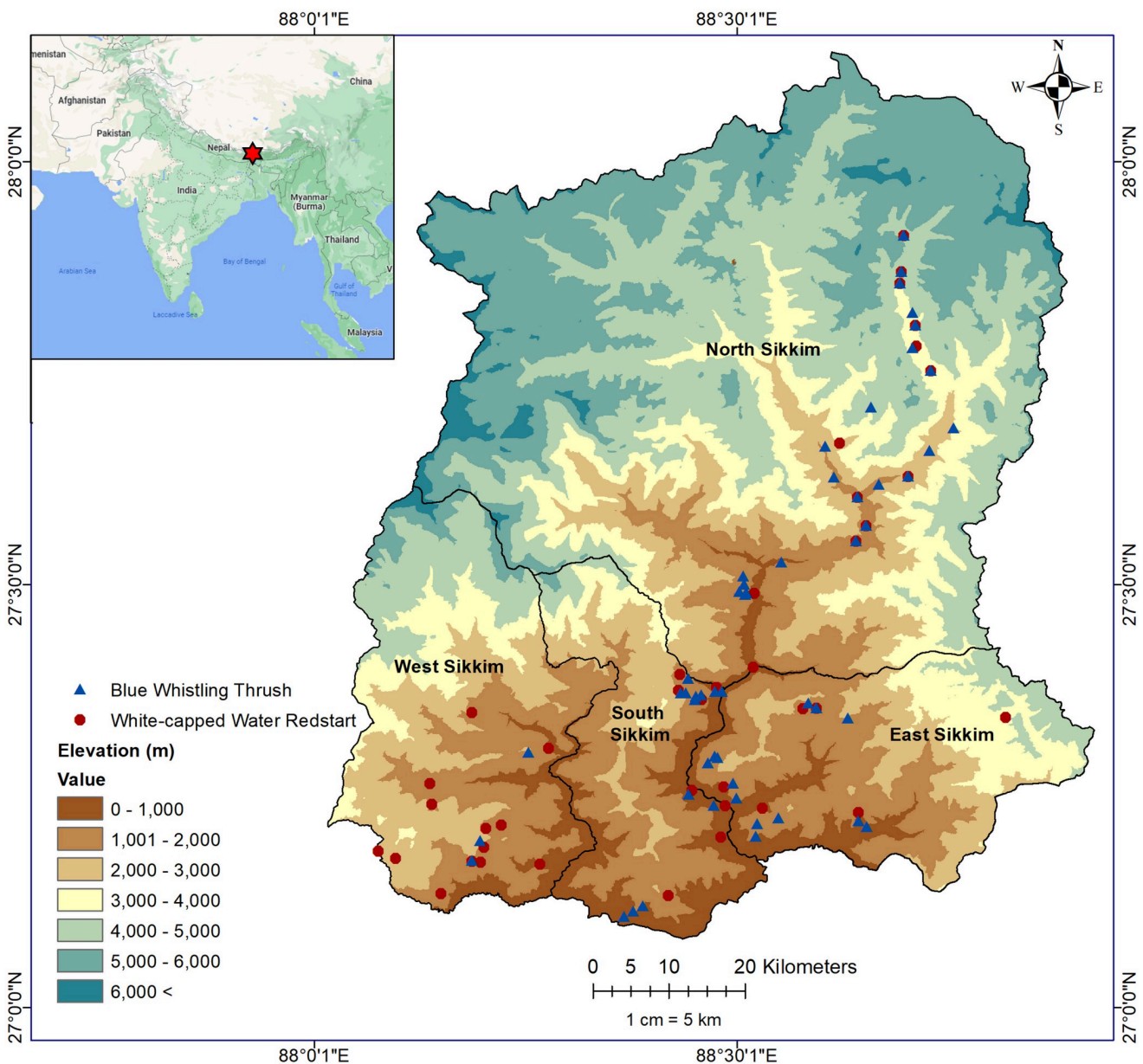

**Fig 1. Study area: Occurrence locations of both study species within Sikkim (inset: Red mark shows the location of Sikkim in India).**

'distance to water' layers in the analysis. We extracted ASTER global digital elevation model (DEM) version 3 (https://earth.data.nasa.gov/) for the elevation layer. Raster files of aspect and slope were created from this DEM using the 'spatial analyst tool' in ArcGIS (version 10.8). For LULC layer we downloaded Sentinel-2 10m time series data available at raster format and having 11 LULC classes (https://www.arcgis.com/). We downloaded a vector layer of the river network from the freely available database HydroSHEDS (https://www.hydrosheds.org/). 'Distance to water' raster was created from this vector layer using the 'Euclidean distance' tool in ArcGIS [32]. We initially considered a total of 23 variables including 19 bioclimatic layers available in Chelsea datasets and four topographic layers (elevation, slope, land-use land-cover

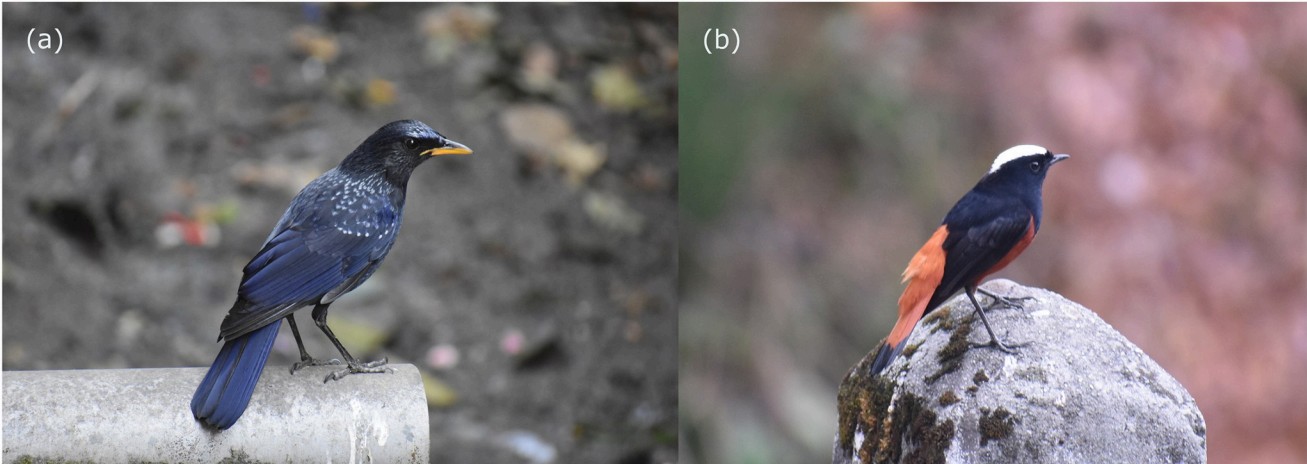

**Fig 2. Images of the study species: (a) Blue Whistling Thrush, and (b) White-capped Water Redstart.**

and distance to water) for the analysis. We made a subset of variables which are ecologically meaningful for the distribution of both study species. We then checked for multicollinearity among the predictor variables and selected only those variables with pairwise Pearson's correlation coefficient $|r| < 0.7$ for the modelling [33]. We used the 'vifcor' function in the R statistical package 'usdm' to perform this step. Here, the variables having higher variance inflation factor (VIF) were excluded. Out of the 23 layers considered initially, we retained only six variables for the final model building which included three bioclimatic variables (bio1- mean annual temperature, bio12-mean annual precipitation, bio15-precipitation seasonality) and three topographic variables (slope, land-use land-cover and distance to water). To reduce the effect of spatial autocorrelation among the occurrence locations, we removed the spatially aggregated occurrence records by keeping a 2 km buffer distance. Hence, we used only 35 records of BWT (out of 52) and 36 records of WWR (out of 40) for the final model building.

## Ecological niche modelling

We created ensemble species distribution models (SDM) using the 'sdm' package in the R statistical programme (version 4.3.1). We used five different algorithms for the model with appropriate pseudo-absence selection, following Barbet-Massin et al. (2012) [34]. We included a combination of both machine-learning and regression based algorithms such as MaxEnt; Multivariate Adaptive Regression Splines (MARS); Multiple Discriminant Analysis (MDA), Random Forest (RF) and Boosted Regression Trees (BRT). We implemented standard parameterization from the 'sdm' package with ten runs of a four-fold cross validation technique [35]. We calculated true skill statistics (TSS) and the area under the receiver operating characteristic (AUC) for each run. We then used a 'weighted' method to develop an ensemble of all the models [36].

## Assessment of habitat characteristics for conservation

We categorized the potential distribution range of both species based on the elevation and coverage of the protected area (PA) network to prioritize conservation actions. For this, we first generated a binary file of the ensemble output using the 'evaluates' function in the 'sdm' package where the threshold value of the maximised TSS was selected [36]. We categorized the

DEM of Sikkim into seven zones at each 1000 m elevation as major habitat transitions are observed at 1000 m intervals in the study region [45]. We overlaid the DEM with the presence map (binary file) and calculated the area of suitable habitats under each elevation category to understand important zones for implementing habitat conservation.

For PA coverage analysis, we downloaded the shapefiles of the PA network from the World Database on Protected Areas (https://www.protectedplanet.net). The boundary of Kitam Bird Sanctuary (which was not included in the WDPA database) was collected from Sikkim Forest Department. We then calculated the coverage of the Protected Area Network by quantifying their overlap with suitable habitats of each species. For this, we overlaid the PA shapefile with the distribution map (binary file) derived from the ensemble model and identified suitable habitats of both bird species covered by PA networks in Sikkim.

## Results

### Model performance

We predicted suitable habitats for BWT and WWR in the Sikkim Himalayan region using ensemble distribution modelling approach. Our models for both species performed well across the evaluation matrices. The average AUC value for BWT was 0.908 and WWR was 0.913 indicating the high performance of the model [35].

### Habitat suitability and variable contributions

For both species, the most suitable habitats are distributed along the major drainage systems which comprise Teesta, Rangeet and their tributaries (Fig 3). Based on the binary map, BWT has 1565.5 km$^2$ area as highly suitable habitats and WWR has 1697.8 km$^2$ suitable habitat area (Fig 4). Interestingly, both species share 1226.3 km$^2$ area of their highly suitable habitats.

Among the predictor variables, annual mean temperature (bio1) had the highest contribution to the distribution models of both bird species. For BWT, annual mean temperature (bio 1) was the major predictor variable with a 54.8% relative contribution followed by mean annual precipitation (bio 12) with a 19.94% and bio15 with 19.9% contribution. In the case of WWR, bio1 had 64.7% relative contribution followed by mean annual precipitation with 22.4% contribution (Fig 5). Overall, the bioclimatic variables had significant contributions compared to the topographic variables.

### Distribution of suitable habitats and PA coverage

Both species have potential habitats across wide elevation ranges (500–6000 m asl), however, the distribution of the most suitable habitats are concentrated in lower to middle elevations (up to 3000 m asl) along the major river drainage systems (Fig 4). About 70 percent of the suitable habitats for both species are distributed within 2000 m ASL of the study area (Table 1).

Out of the most suitable habitats for BWT and WWR, only eight percent of the area is covered by the protected area network (Fig 6).

## Discussion

Species distribution modelling approaches are widely used in formulating habitat conservation strategies across the globe. Our study is the first of its kind from the Eastern Himalaya by taking two common water-dependent birds as indicator species to address important conservation needs of riverine habitats. Moreover, understanding the distribution of currently common species will help us to protect them from future risks of habitat loss and population declines within the region. The Himalayan biodiversity hotspot is reported to have a high

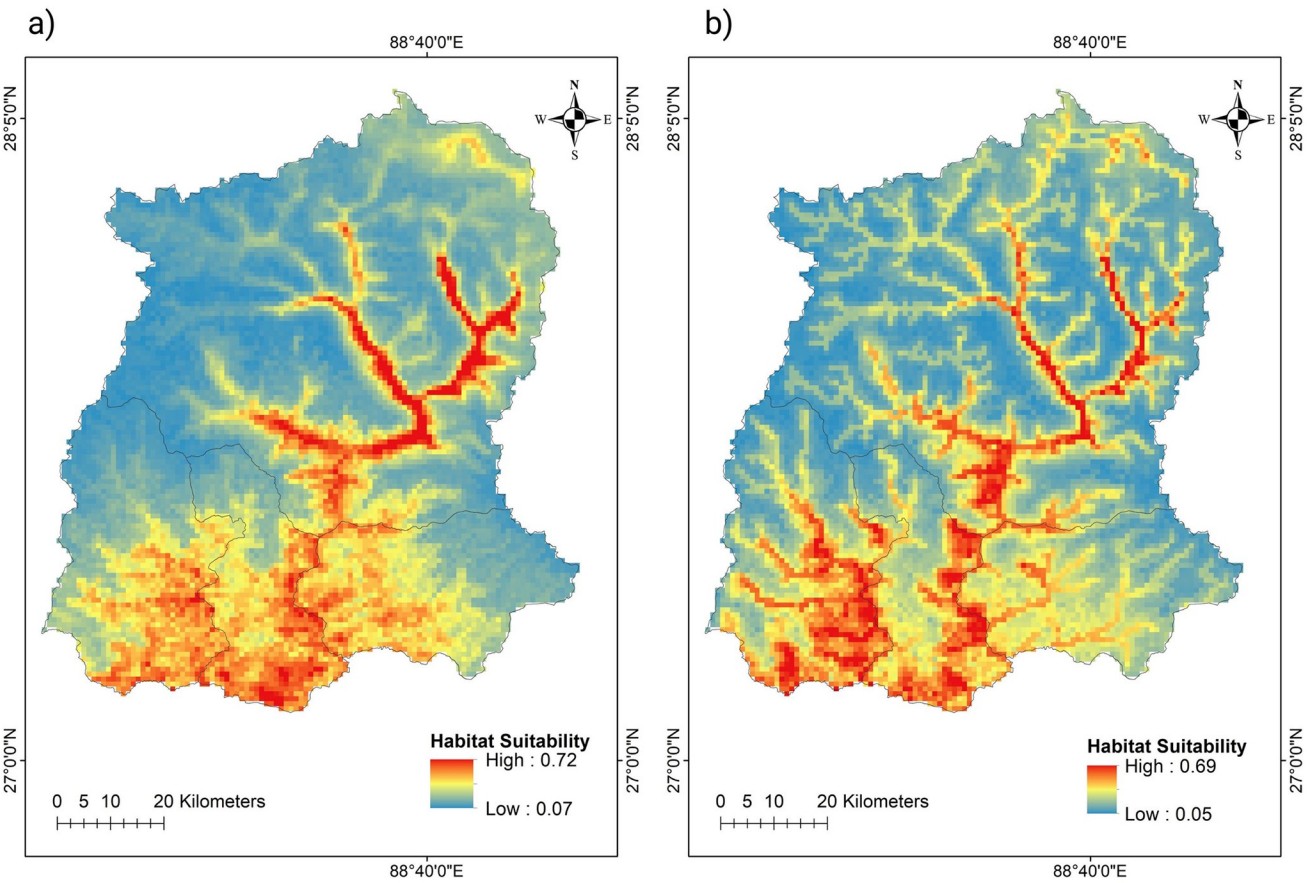

**Fig 3. Suitable habitats of (a) Blue Whistling Thrush and (b) White-capped Water Redstart in the Sikkim Himalaya.** The values range from 0 to 1 where the area with value 1 has the highest chance for the species to be found.

diversity of specialist river birds compared to any other region [17]. However, studies focusing on habitat suitability and conservation strategies of riverine birds are rare in this region. Observations from Western Himalaya highlighted that the impact of anthropogenic disturbances on riverine habitats is significant where the river birds can act as excellent indicators of these changes [15].

Here we predicted suitable habitats for two water-dependent birds namely, Blue Whistling Thrush and White-capped Water Redstart in the Sikkim Himalaya and derived conservation implications for riverine ecosystems. Our models performed well and revealed some interesting patterns in the distribution of suitable habitats for these riverine birds. Overall, the outputs of the distribution model indicated that the suitable habitats of these birds ideally represent the riverine ecosystems of the Sikkim Himalayan region which is of immediate conservation concern.

Our results highlighted that the bioclimatic variables are more important for both species compared to the topographic variables. Annual mean temperature (bio1) was the most important factor for both study species indicating the importance of temperature patterns for their distribution. Both species are reported to be exhibiting local migratory behaviour along elevation gradients during the breeding seasons [37] hence, the changes in climatic conditions including temperature may be critical for their survival. Environmental temperature is one of

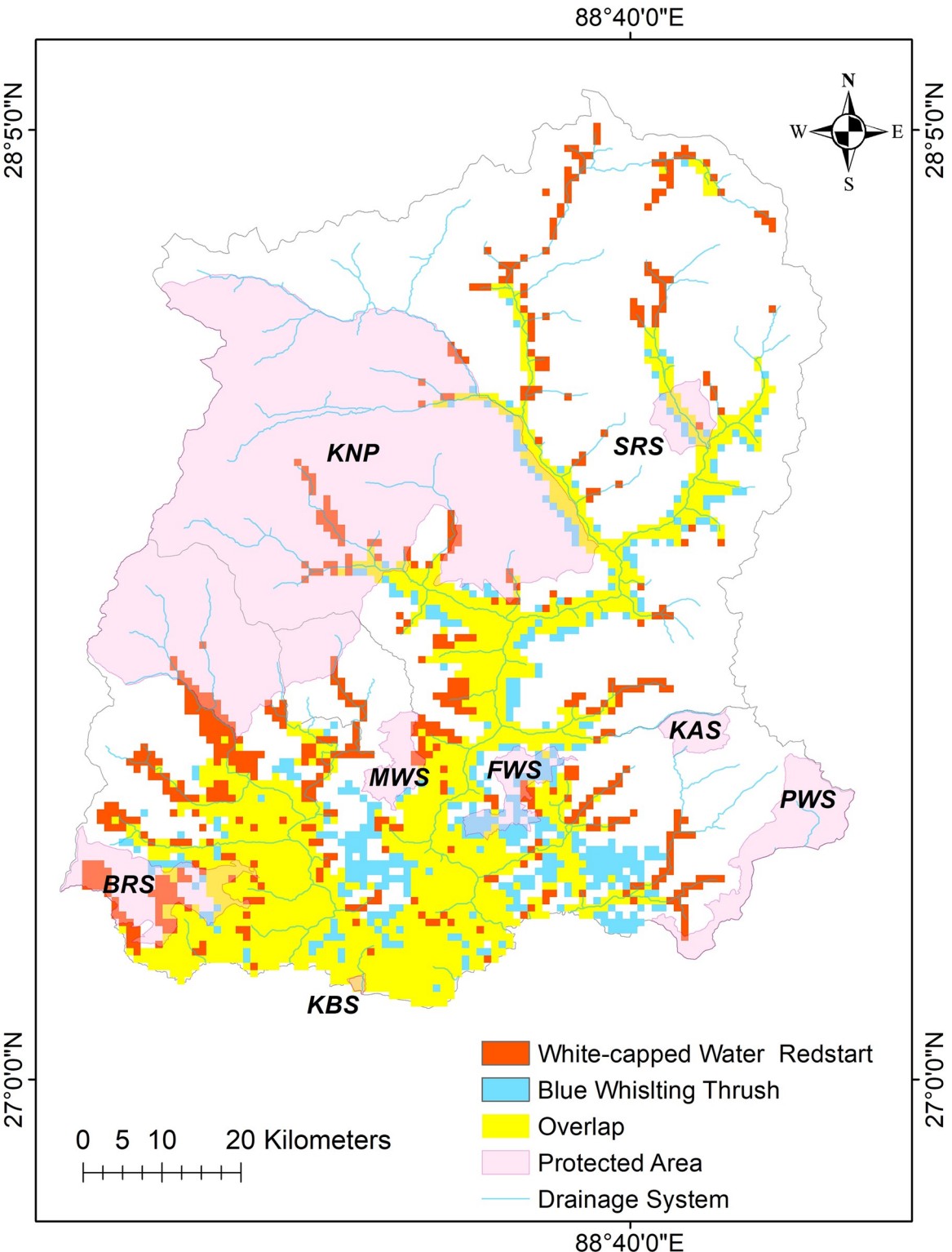

**Fig 4. Distribution of highly suitable habitats of both study species across Sikkim Himalaya along with protected area network and major drainage systems.** SRS-Shingba Rhododendron Sanctuary, KNP-Khangchendzonga National Park, KAS-Kyongnosla Alpine Sanctuary, FLS- Fambong Lho Wildlife Sanctuary, MWS- Maenam Wildlife Sanctuary, BRS-Barsey Rhododendron Sanctuary, PWS-Pangolakha Wildlife Sanctuary, KBS- Kitam Bird Sanctuary.

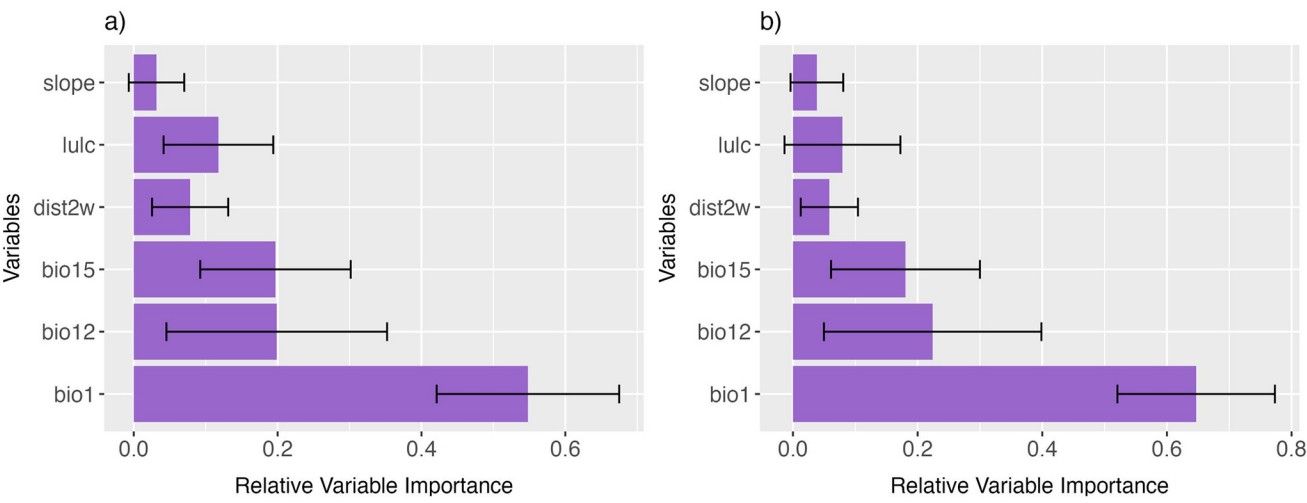

**Fig 5. Relative importance of predictor variables in the final model: (a) Blue Whistling Thrush and (b) White-capped Water Redstart.** bio1- mean annual temperature, bio12-Mean annual precipitation, bio15-Precipitation seasonality, lulc-Land-use and land-cover, dist2w- Distance to water.

the major constraints for bird distribution in any ecosystem [38]. Also, temperature regimes are reported to be an important factor in directly or indirectly regulating elevational distributions of species on mountains [39]. A study on the breeding behaviour of Eastern Himalayan birds highlighted that many birds including the common and wide-ranged species exhibit elevational preferences for their breeding grounds [37]. The commonly found waterbirds including BWT and WWR were observed only from select elevation zones during their breeding seasons. Such findings indicated that even common species have specific habitat requirements for breeding irrespective of their wider elevation distribution ranges. Following the temperature, higher contribution of bio12 (mean annual precipitation) and bio15 (precipitation seasonality) indicated that these birds could be highly sensitive to moisture levels. Studies from the region observed changes in the temporal distribution of rainfall with a drastic decline in winter rains [40]. The rainfall patterns have a significant impact on riverine ecosystems in the Himalaya [41]. Since these species are mostly found near rivers and streams, the availability of moist habitats, especially during dry seasons, could be very important for the survival of the birds.

Our results also revealed that a large proportion of the most suitable habitats of these birds are in the low to mid-elevation especially within 2000 m asl. Both species have a wider

**Table 1. High suitability areas of both bird species in each elevation zone in the Sikkim Himalaya.** The value within the bracket shows the percentage with respect to total suitable area.

| Sl. | Elevation zone (m) | Suitable habitat area (km$^2$) | |
|---|---|---|---|
| | | **Blue Whistling Thrush** | **White-capped Water Redstart** |
| 1 | 0–1000 | 399.16 (125.50%) | 393.71 (23.19%) |
| 2 | 1000–2000 | 838.78 (53.58%) | 818.55 (48.21%) |
| 3 | 2000–3000 | 190.63 (12.18%) | 281.67 (16.59%) |
| 4 | 3000–4000 | 111.27 (7.11%) | 117.49 (6.92%) |
| 5 | 4000–5000 | 17.12 (1.09%) | 52.13 (3.07%) |
| 6 | 5000–6000 | 8.56 (0.55%) | 34.24 (2.02%) |
| | **Total** | 1565.51 | 1697.78 |

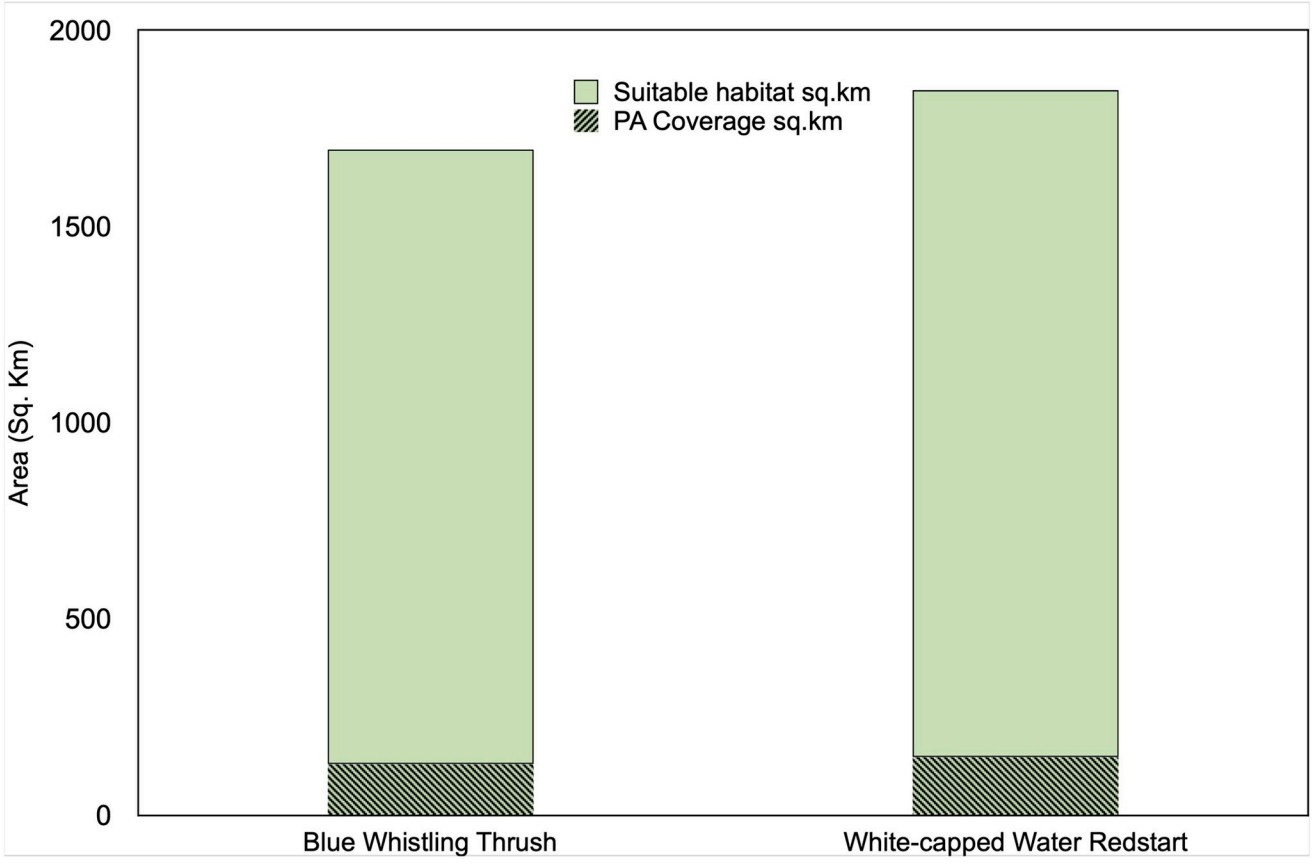

**Fig 6. Highly suitable habitats and protected area coverage for both study species in Sikkim Himalaya.**

(horizontal) distribution range in the lower elevations than in higher elevations, and this could be because of the availability of more areas with a relatively stable temperature in lower zones. Thermal tolerances are reported to regulate bird distributions across different locations and elevation zones [42].

Sikkim has eight protected areas within a small geographical range covering more than 30% of its geographical area, however, the lower elevation zones are mostly outside these networks [13]. These zones are dominated by human settlements and are under the pressure of anthropogenic activities. Our analysis showed that the majority of suitable habitats for both bird species fall outside Protected Area Network, especially in the lower elevations. Including our study species, other birds such as Plumbeous Water Redstart, Brown Dipper, Forktails, Wagtails and many other water-dependent birds share the same habitat within the Himalayan range [43, 44]. A study on freshwater fish diversity from the Western Ghats mountains of India highlighted that terrestrial protected areas are often not sufficient to protect riverine habitats and we need additional efforts in conservation to protect these ecosystems [40]. Gap analyses by integrating predicted species ranges and levels of protection or disturbances are important to expand riverine ecosystem conservation [45]. Hence, a quality assessment of riverine ecosystems including the potential habitats of riverine birds outside the PA network in the Eastern Himalaya is critical.

The rapid reduction of glaciers- one of the most reported impacts of climate change- in the Himalaya can largely influence downstream water supplies [46]. The potential impacts of

climate change, specifically on water birds and riverine ecosystems, are highlighted in some of the studies from the Himalaya. For example, it is reported that changes in surface temperatures of water bodies can reduce prey availability for river birds [47]. The changes in climate and land-use patterns in the Himalaya have resulted in deterioration of the water quality and quantity, biodiversity changes, and the decreasing migratory bird population in the wetlands [48]. Moreover, various urbanization activities and habitat modifications are influencing the distribution of river birds in the Himalaya [15]. Rapid developmental activities including hydroelectric projects and pharmaceutical companies are reported to alter riverine habitats in the Eastern Himalaya. In this scenario, it is important to protect the remaining suitable habitats of these species in the region before the current changes in land-use and climate could lead them to local extinctions.

We recommend a systematic and detailed habitat assessment of water-dependent birds in the Sikkim Himalayan region to plan immediate conservation strategies. An ecosystem approach will benefit multiple species of river birds along with other water-dependent taxa such as odonates, fishes and herpetofauna in the Eastern Himalaya.

## Conclusion

Our study provided insights into the requirements of critical habitat management strategies for Himalayan riverine habitats- one of the unique riverine ecosystems in the world. Although we used commonly found water birds for our study, the results raise important concerns about the riverine ecosystems in the Eastern Himalaya. Since these birds are relatively common, we managed to collect a good number of occurrence records for the model with increased performance. These species acted as representatives of the riverine communities as cost-effective indicators of riverine ecosystems in the region. Our results also reveal that irrespective of their global status as 'least concern', these species may soon lose their regional habitats if we do not implement timely conservation actions, especially outside PA networks. These ecosystems within the Himalayan biodiversity hotspot are of global importance, however, undergoing large-scale changes in land-use and climatic conditions. Disturbances and changes in these riverine habitats will not only threaten the study species but also result in population decline and local extinction of several other riverine birds in the region. Ground-level action plans and policies are recommended for regulating developmental activities in the river basins of the Eastern Himalaya for the better conservation of riverine ecosystems and associated biodiversity.

## Supporting information

**S1 Table. Details of sighting records of Blue Whistling Thrush in different parts of Sikkim Himalaya.**
(DOC)

**S2 Table. Details of sighting records of White-capped water redstart obtained in different parts of Sikkim.** (*) Data obtained from others.
(DOC)

## Acknowledgments

We are grateful to Sikkim University for the research facilities to undertake field and laboratory work. We thank the State Forest Department of Sikkim for permissions to carry out field surveys. We thank Luíz Fernando Esser, Anoop NR and Abhishek Samrat for their inputs and

suggestions in the analysis. We also acknowledge the help and support received from field assistants and local communities for conducting fieldwork in the Sikkim Himalaya.

## Author Contributions

**Conceptualization:** Shivaji Chaudhry, Seema Rawat, Bhoj Kumar Acharya.

**Data curation:** Roshan Tamang, Sailendra Dewan, Bhoj Kumar Acharya.

**Formal analysis:** Roshan Tamang, Vallanattu James Jins, Sailendra Dewan.

**Funding acquisition:** Bhoj Kumar Acharya.

**Investigation:** Roshan Tamang, Sailendra Dewan.

**Methodology:** Vallanattu James Jins, Sailendra Dewan.

**Software:** Roshan Tamang, Vallanattu James Jins, Sailendra Dewan.

**Supervision:** Bhoj Kumar Acharya.

**Writing – original draft:** Roshan Tamang, Vallanattu James Jins, Sailendra Dewan.

**Writing – review & editing:** Shivaji Chaudhry, Seema Rawat, Bhoj Kumar Acharya.

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
