## [Decision Letter · Decision Letter 0]

29 Mar 2023

PONE-D-23-07258Ecological niche modelling of two water-dependant birds informs conservation needs of riverine ecosystems outside protected area network in the Eastern Himalaya, IndiaPLOS ONE

Dear Dr. Acharya,

Thank you for submitting your manuscript to PLOS ONE. After careful consideration, we feel that it has merit but does not fully meet PLOS ONE’s publication criteria as it currently stands. Therefore, we invite you to submit a revised version of the manuscript that addresses the points raised during the review process.

We look forward to receiving your revised manuscript.

Kind regards,

Maharaj K Pandit, Ph.D.

Academic Editor

PLOS ONE

Journal Requirements:

2. We note that Figure 1, 3, and 4 in your submission contain copyrighted images. All PLOS content is published under the Creative Commons Attribution License (CC BY 4.0), which means that the manuscript, images, and Supporting Information files will be freely available online, and any third party is permitted to access, download, copy, distribute, and use these materials in any way, even commercially, with proper attribution. For more information, see our copyright guidelines: http://journals.plos.org/plosone/s/licenses-and-copyright.

1. You may seek permission from the original copyright holder of  Figure 1, 3, and 4 to publish the content specifically under the CC BY 4.0 license. 

Reviewers' comments:

Reviewer's Responses to Questions

**Comments to the Author**

1. Is the manuscript technically sound, and do the data support the conclusions?

Reviewer #1: No

2. Has the statistical analysis been performed appropriately and rigorously? 

Reviewer #1: No

3. Have the authors made all data underlying the findings in their manuscript fully available?

Reviewer #1: Yes

4. Is the manuscript presented in an intelligible fashion and written in standard English?

Reviewer #1: Yes

5. Review Comments to the Author

Reviewer #1: The manuscript entitled “Ecological niche modelling of two water-dependant birds informs conservation needs of riverine ecosystems outside protected area network in the Eastern Himalaya, India” is an interesting study but suffer from some major methodological flaws. Hence, I would recommend “Major Revision” at this stage. The authors should try to improve their manuscript as per the following suggestions:

1. As far as I understand, the authors have outlined “distance to water” to be the main factor for determining species presence of Blue Whistling Thrush and White-capped Water Redstart. The authors report that “distance to water” was calculated as Euclidean distance from the vector file of drainage systems of the DEM of the study area. If this is the case, then wouldn’t the resolution of DEM determine the output of “distance to water”? Also, the resolution of DEM and resolution of sampling effort does not match (If i am not wrong)? As per my understanding, “flow distance” is a better measure for measuring geographic distances in river networks. The reason being that Euclidean distance represents shortest straight line distance while flow distance joined by river flow. So, Euclidean distance might misrepresent the distance to river as it may indicate those sites that actually do not have any river water. The authors should consider this fact.

2. My second major concern is with respect to determining conservation needs (as reflected in the title) for bird species that themselves fall in the “Least Concern” Category of Red List (Blue Whistling Thrush and White-capped Water Redstart). The study area (Sikkim) is a hot-spot of avifaunal diversity with many endemic and threatened species. Why determine conservation priorities based on the most common species?

3. Another major flaw is that the conservation priorities have been determined based on just geographic and climatic factors. Wouldn’t it be more prudent to use “forest cover classes” as one of the predictor in SDM analyses as the study is with respect to birds?

4. I am not sure if use of only one SDM technique (MaxEnt) is good for predicting habitat suitability for small datasets. It would be worthwhile to use “ensemble modelling approach” for determining habitat suitability. For example, see the work of Ahmad et al. (2020) who used ensemble modelling approach for predicting habitat suitability of Himalayan goral.

5. The authors state that the protected area boundaries were sourced from World database of protected areas (WDPA). WDPA database does not include “Kitam Bird Sanctuary”. Was this PA included in the study or not? Also, Figure 3 depicts incorrect boundary of “Kyongnosla Alpine Wildlife Sanctuary”. Please cross-check. Also, name the individual PA in Figure 3.

6. In Table 1, habitat suitability values for Elevation zone 6000-7000 m have been given for two bird species. Are these areas potential habitats for these bird species?

7. Lines 98-106: The authors mention in detail the various kinds of anthropogenic pressures threatening the Sikkim Himalayan ecosystems? What is the use if no such anthropogenic threat is included in the analyses of suitable habitats?

8. Would dam-building threaten the habitat of the two bird species (given the fact the Sikkim has one of the highest dam densities in the Himalaya and in the study the authors have concluded “distance to water” as the main predictor of species presence). Again, here I feel that if the authors had included some other bird species that ranks higher in IUCN red list, the study would had become very interesting.

6. PLOS authors have the option to publish the peer review history of their article (what does this mean?). If published, this will include your full peer review and any attached files.

Reviewer #1: No

---

## [Author Response · Author response to Decision Letter 0]

15 Jul 2023

Response to Reviewers Comments

PONE-D-23-07258

Ecological niche modelling of two water-dependant birds informs conservation needs of riverine ecosystems outside protected area network in the Eastern Himalaya, India

General Comment: The manuscript entitled “Ecological niche modelling of two water-dependant birds informs conservation needs of riverine ecosystems outside protected area network in the Eastern Himalaya, India” is an interesting study but suffer from some major methodological flaws. Hence, I would recommend “Major Revision” at this stage. The authors should try to improve their manuscript as per the following suggestions:

Comment 1. As far as I understand, the authors have outlined “distance to water” to be the main factor for determining species presence of Blue Whistling Thrush and White-capped Water Redstart. The authors report that “distance to water” was calculated as Euclidean distance from the vector file of drainage systems of the DEM of the study area. If this is the case, then wouldn’t the resolution of DEM determine the output of “distance to water”? Also, the resolution of DEM and resolution of sampling effort does not match (If I am not wrong)? As per my understanding, “flow distance” is a better measure for measuring geographic distances in river networks. The reason being that Euclidean distance represents shortest straight line distance while flow distance joined by river flow. So, Euclidean distance might misrepresent the distance to river as it may indicate those sites that actually do not have any river water. The authors should consider this fact.

Author response: Yes, our results showed that ‘distance to water’ is more important for White-capped water redstart than Whistling thrush. We have revised the analyses based on reviewer comments. We have now used the data downloaded from HydroSHEDS which has high resolution data of global river works. This is a vector file representing the actual river flow. We created raster using euclidean distance tool in the same resolution (1x1 km) as our samples and pother predictor variables. This is a standard method for species distribution modeling approaches involving ‘distance to water’ layers. For example Mothes et al 2020 used the same method to model distribution of imperiled wood turtle (a water-dependant species) in the North America. We have cited this paper for the method we used in our analysis.

Comment 2. My second major concern is with respect to determining conservation needs (as reflected in the title) for bird species that themselves fall in the “Least Concern” Category of Red List (Blue Whistling Thrush and White-capped Water Redstart). The study area (Sikkim) is a hot-spot of avifaunal diversity with many endemic and threatened species. Why determine conservation priorities based on the most common species?

Author response: We agree with the reviewer in this context as conservation actions till now are mostly focused on rare and at-risk species. However, common species also have their own ecological and functional roles in ecosystems. Conservation of such species is critical to prevent them from rapid decline and local extinctions. Although these species are ‘least concern’ they might have specific ecological and habitat requirements which make them vulnerable at local scales. Also, we used these common species as indicator species of riverine ecosystems where they share space with many other sympatric species. The importance of addressing the conservation needs of common species are highlighted in Lindenmayer et al. 2011 (cited). We have made necessary changes in the revised manuscript now highlighting these issues

Comment 3. Another major flaw is that the conservation priorities have been determined based on just geographic and climatic factors. Wouldn’t it be more prudent to use “forest cover classes” as one of the predictor in SDM analyses as the study is with respect to birds?

Author response: Thank you for the suggestion. In the revised version we included land use and land cover layer (which also represents forest classes) as one of the predictor variables along with other predictor variables. However, this variable hasn’t turned out to be an important factor for the distribution of both species.

Comment 4. I am not sure if use of only one SDM technique (MaxEnt) is good for predicting habitat suitability for small datasets. It would be worthwhile to use “ensemble modelling approach” for determining habitat suitability. For example, see the work of Ahmad et al. (2020) who used ensemble modelling approach for predicting habitat suitability of Himalayan goral.

Author Response: Among many SDM approaches MaxEnt is reported to be one of the best method for predicting species with even small number of occurrence records (Pearson 2007).

[Pearson, R. G., Raxworthy, C. J., Nakamura, M., & Townsend Peterson, A. (2007). Predicting species distributions from small numbers of occurrence records: a test case using cryptic geckos in Madagascar. Journal of biogeography, 34(1), 102-117.]

Although several new modeling approaches have been evolved in recent years, MaxEnt is still considered to be among the best ones due to its simplicity and high-performance. Kaky et al 2020 highlighted that single-algorithm modelling methods, particularly MaxEnt, are capable of producing distribution maps of comparable accuracy to ensemble methods.

Further there are studies showing no particular benefit to using ensembles over individual tuned models. 

Examples:

[Kaky, E., Nolan, V., Alatawi, A., & Gilbert, F. (2020). A comparison between Ensemble and MaxEnt species distribution modelling approaches for conservation: A case study with Egyptian medicinal plants. Ecological Informatics, 60, 101150.

Hao, T., Elith, J., Lahoz‐Monfort, J. J., & Guillera‐Arroita, G. (2020). Testing whether ensemble modelling is advantageous for maximising predictive performance of species distribution models. Ecography, 43(4), 549-558.

Comment 5. The authors state that the protected area boundaries were sourced from World database of protected areas (WDPA). WDPA database does not include “Kitam Bird Sanctuary”. Was this PA included in the study or not? Also, Figure 3 depicts incorrect boundary of “Kyongnosla Alpine Wildlife Sanctuary”. Please cross-check. Also, name the individual PA in Figure 3.

Author Response: Thank you for highlighting this issue. We have included the Kitam Bird Sanctuary in the map using the boundary collected from Sikkim Forest Department. This was added to the WDPA data base that we created for the final PA area shape file. We cross-checked with the available data and the boundary of Kyongnosla Alpine Wildlife Sanctuary seems to be correct (https://wiienvis.nic.in/WriteReadData/UserFiles/image/PAs_Map_Database/images/sikkim2.jpg).

Comment 6. In Table 1, habitat suitability values for Elevation zone 6000-7000 m have been given for two bird species. Are these areas potential habitats for these bird species?

Author Response: After the revised analysis, there is only negligible area (less than 1 square km) above 6000m as suitable habitat (only for Blue Whistling Thrush). Although the model shows it as potential habitat, we will need further surveys to confirm their actual presence in these elevations. Considering the extreme weather conditions it is less likely that the species might actually occur there although our model shows some suitability.

Comment 7. Lines 98-106: The authors mention in detail the various kinds of anthropogenic pressures threatening the Sikkim Himalayan ecosystems? What is the use if no such anthropogenic threat is included in the analyses of suitable habitats?

Author Response: In conservation perspective, SDM analysis are mainly used for delineating most suitable habitats than identifying predictor variables. In modelling, we need to use samples and predictor layers in same resolution. Accounting multiple anthropogenic threats as a layer in SDM especially for such complex areas like Himalaya may not be practical. However, once we identify their potential habitats using other variables (such as bioclimatic) we can go further exploring most suitable habitats and understand the threats at local level. Our analysis will help to further investigate the threats in the riverine ecosystems of Sikkim Himalaya. Nonetheless we have highlighted these issues because they are very important from conservation point of view.

Comment 8. Would dam-building threaten the habitat of the two bird species (given the fact the Sikkim has one of the highest dam densities in the Himalaya and in the study the authors have concluded “distance to water” as the main predictor of species presence). Again, here I feel that if the authors had included some other bird species that ranks higher in IUCN red list, the study would had become very interesting.

Author Response: Dam-building is of course a threat to the riverine ecosystems in the region. As mentioned above (in response to comment 2), common species also need attention as they undertake specific ecological and functional roles in respective ecosystems. Moreover, our study species act as indicator species of riverine ecosystem and share habitats with many other important water-dependant species. Hence, our approach may help multiple species if we could develop strategies for these riverine habitats.

---

## [Decision Letter · Decision Letter 1]

1 Aug 2023

PONE-D-23-07258R1Ecological niche modelling of two water-dependant birds informs conservation needs of riverine ecosystems outside protected area network in the Eastern Himalaya, IndiaPLOS ONE

Dear Dr. Acharya,

Thank you for submitting your manuscript to PLOS ONE. After careful consideration, we feel that it has merit but does not fully meet PLOS ONE’s publication criteria as it currently stands. Therefore, we invite you to submit a revised version of the manuscript that addresses the points raised during the review process. Please pay careful attention to the reviewers comments.   

We look forward to receiving your revised manuscript.

Kind regards,

Maharaj K Pandit, Ph.D.

Academic Editor

PLOS ONE

Reviewers' comments:

Reviewer's Responses to Questions

**Comments to the Author**

1. If the authors have adequately addressed your comments raised in a previous round of review and you feel that this manuscript is now acceptable for publication, you may indicate that here to bypass the “Comments to the Author” section, enter your conflict of interest statement in the “Confidential to Editor” section, and submit your "Accept" recommendation.

Reviewer #1: (No Response)

2. Is the manuscript technically sound, and do the data support the conclusions?

Reviewer #1: Partly

3. Has the statistical analysis been performed appropriately and rigorously? 

Reviewer #1: No

4. Have the authors made all data underlying the findings in their manuscript fully available?

Reviewer #1: No

5. Is the manuscript presented in an intelligible fashion and written in standard English?

Reviewer #1: Yes

6. Review Comments to the Author

Reviewer #1: I have now completed the review of the revised manuscript entitled "Ecological niche modelling of two water-dependant birds informs conservation needs of riverine ecosystems outside protected area network in the Eastern Himalaya, India". While I find that the authors have addressed some of my comments from the previous review, I find that they have still not addressed my two major comments:

(i) They have still used only two most "common species" namely "Blue Whistling Thrush" and "White-capped Water Redstart" for determining conservation prioritization. In my previous review, I had highlighted the fact that conservation prioritization only works best when it is done for IUCN Red List species or Endemic species. I will strongly request the authors to redo SDM analyses by including IUCN Red List Species of Sikkim Himalaya and then do conservation prioritization exercise. Right now, the results hold no significance as the two subject species (Blue Whistling Thrush and White Capped Water Redstart) are prevalent everywhere in the study area irrespective of the threats. In a nutshell, they do not serve as good ecological indicators for threats. Hence, it would be best for the authors to gather species presence for the IUCN Red List species of Sikkim through primary/secondary databases and then perform conservation prioritization exercise.

(ii) The authors have still not used "ensemble modelling approach". I had highlighted in the previous review also that databases like Chelsa climate database that the authors have used are of global resolution. As such SDMs work best when local level climate models are used. I know that such databases do not exist for the Himalaya or Sikkim Himalayan region. Therefore, in these cases it is best to use ensemble modelling approach where climate simulations from different global databases are used and then the mean value is taken as a representation of the habitat/climate of the study area. This process removes any inherent bias with respect to downscaling of data from the global to the regional levels. Therefore, I will again suggest the authors to use different global climate model databases to perform SDM exercise and then take the mean values.

The above are the two major comments that i think needs to be definitely addressed before potential publication.

7. PLOS authors have the option to publish the peer review history of their article (what does this mean?). If published, this will include your full peer review and any attached files.

Reviewer #1: No

---

## [Author Response · Author response to Decision Letter 1]

9 Oct 2023

Dear Editor,

We thank you for providing us a constructive review for our manuscript submitted to PLoS ONE. We have revised our manuscript based on reviewer’s comments and incorporated changes accordingly. There were two major concern of the reviewers (1) to change the model species from common birds to IUCN red listed species, and (2) to change modelling approach from MaxEnt to ensemble. I would like to mention that we have fully addressed concern 2 and reanalysed our model (ensemble) and addressed all the concerns raised by reviewer. Regarding the change of model bird species, we have provided strong justification below for why we cannot change the species from common to IUCN red listed. Almost all researchers focus only on IUCN red listed species but these common species are equally important from conservation point of view and are neglected by researchers and conservationists. Hence, we stick to the common species because our entire aim of this MS will be diluted if we change the model species . We believe that our manuscript is now in a better shape to consider for publication in your esteemed journal.

Review Comments to the Author

Reviewer #1: I have now completed the review of the revised manuscript entitled "Ecological niche modelling of two water-dependant birds informs conservation needs of riverine ecosystems outside protected area network in the Eastern Himalaya, India". While I find that the authors have addressed some of my comments from the previous review, I find that they have still not addressed my two major comments:

(i) They have still used only two most "common species" namely "Blue Whistling Thrush" and "White-capped Water Redstart" for determining conservation prioritization. In my previous review, I had highlighted the fact that conservation prioritization only works best when it is done for IUCN Red List species or Endemic species. I will strongly request the authors to redo SDM analyses by including IUCN Red List Species of Sikkim Himalaya and then do conservation prioritization exercise. Right now, the results hold no significance as the two subject species (Blue Whistling Thrush and White Capped Water Redstart) are prevalent everywhere in the study area irrespective of the threats. In a nutshell, they do not serve as good ecological indicators for threats. Hence, it would be best for the authors to gather species presence for the IUCN Red List species of Sikkim through primary/secondary databases and then perform conservation prioritization exercise.

We thank the reviewer for this comment but as we highlighted as part of the reply to first review comments, we would like to mention that our study is using these common species as indicators of riverine ecosystems. The main aim of this paper itself is to look for distribution and threats experienced by common species because most study focus only on IUCN listed species and these currently common species are always ignored, and not taken into consideration. Common species often play vital roles in ecosystem functions and processes. Globally, conservation strategies are mostly focused on threatened species and rarely explored the potential of using common species as indicators of critical ecosystems. Conserving common species is critical in retaining their key ecological and functional roles in ecosystems. However, conservation strategies are often focused only on threatened species and we need a holistic approach to preserve multiple ecosystem functions. For instance, a habitat conservation approach focusing on the riverine ecosystem could benefit multiple species including sympatric birds and other water-dependent taxa such as dragonflies, damselflies, fishes, amphibians and reptiles. Such approaches are needed to inform local and regional decision-makers on conservation or management options that anticipate the response of biodiversity to future climate and land use change (Please see the introduction part for rationale for using common species). 

There are studies from the Himalayan region (cited in introduction) highlighting these water birds as cost-effective indicators of riverine habitats. Including more species especially threatened ones is challenging as we don’t have sufficient validated number of occurrences from the region. Additionally, the entire aim of the MS will change if we change the model species. Moreover, the region has not many IUCN threatened water-dependent species to include in the model. We used these common species to highlight their specialised niches although they are not threatened currently. However, we feel that such common species and other sympatric species might also have specific habitat requirements that needs to be preserved to protect them from future declines or local extinctions. We made modifications in the text also to highlight the importance of common species and also using them as indicators. We consider our study address multiple species in the ecosystem rather than just addressing conservation of these two target species. In this way we believe that the work is still relevant. Hence, we stick to the common species because our entire aim of this MS will be diluted if we change the model species.

(ii) The authors have still not used "ensemble modelling approach". I had highlighted in the previous review also that databases like Chelsa climate database that the authors have used are of global resolution. As such SDMs work best when local level climate models are used. I know that such databases do not exist for the Himalaya or Sikkim Himalayan region. Therefore, in these cases it is best to use ensemble modelling approach where climate simulations from different global databases are used and then the mean value is taken as a representation of the habitat/climate of the study area. This process removes any inherent bias with respect to downscaling of data from the global to the regional levels. Therefore, I will again suggest the authors to use different global climate model databases to perform SDM exercise and then take the mean values.

Thanks for your suggestion. We have now revised our analysis using ‘ensemble’ approach. We used five algorithms (combination of machine-learning and regression models) to generate combined output of the model. We hope that the model is now more robust. 

Note: Ensemble approaches are recommended for avoiding the bias of using single algorithm but the same time there are studies highlighting that individually tuned models (like Maxent with customised feature settings) are very efficient in many cases where we have less number or spatially aggregated occurences as we highlighted in first revision. 

Regarding Chelsa- studies show that the climate datasets from Chelsa is reported to be more efficient for modeling studies in mountain environments like the Himalaya (Reference: Maria B, Udo S. Why input matters: Selection of climate data sets for modelling the potential distribution of a treeline species in the Himalayan region. Ecol Model. 2017;359: 92–102. doi:10.1016/j.ecolmodel.2017.05.021).

We once again thank the editor and reviewer for providing valuable inputs to the MS which helped us to revise it significantly.

---

## [Decision Letter · Decision Letter 2]

25 Oct 2023

Ecological niche modelling of two water-dependant birds informs conservation needs of riverine ecosystems outside protected area network in the Eastern Himalaya, India

PONE-D-23-07258R2

Dear Dr. Acharya,

We’re pleased to inform you that your manuscript has been judged scientifically suitable for publication and will be formally accepted for publication once it meets all outstanding technical requirements.

Kind regards,

Maharaj K Pandit, Ph.D.

Academic Editor

PLOS ONE

Additional Editor Comments (optional):

Reviewers' comments:

Reviewer's Responses to Questions

**Comments to the Author**

1. If the authors have adequately addressed your comments raised in a previous round of review and you feel that this manuscript is now acceptable for publication, you may indicate that here to bypass the “Comments to the Author” section, enter your conflict of interest statement in the “Confidential to Editor” section, and submit your "Accept" recommendation.

Reviewer #1: All comments have been addressed

2. Is the manuscript technically sound, and do the data support the conclusions?

Reviewer #1: Yes

3. Has the statistical analysis been performed appropriately and rigorously? 

Reviewer #1: Yes

4. Have the authors made all data underlying the findings in their manuscript fully available?

Reviewer #1: Yes

5. Is the manuscript presented in an intelligible fashion and written in standard English?

Reviewer #1: Yes

6. Review Comments to the Author

Reviewer #1: (No Response)

7. PLOS authors have the option to publish the peer review history of their article (what does this mean?). If published, this will include your full peer review and any attached files.

Reviewer #1: No

---

## [Editor Report · Acceptance letter]

31 Oct 2023

PONE-D-23-07258R2 

Ecological niche modelling of two water-dependant birds informs the conservation needs of riverine ecosystems outside protected area network in the Eastern Himalaya, India 

Dear Dr. Acharya:

I'm pleased to inform you that your manuscript has been deemed suitable for publication in PLOS ONE. Congratulations! Your manuscript is now with our production department. 

Kind regards, 

on behalf of

Professor Maharaj K Pandit 

Academic Editor

PLOS ONE